# Ecological Specialization and Rarity of Arable Weeds: Insights from a Comprehensive Survey in France

**DOI:** 10.3390/plants9070824

**Published:** 2020-06-30

**Authors:** François Munoz, Guillaume Fried, Laura Armengot, Bérenger Bourgeois, Vincent Bretagnolle, Joël Chadoeuf, Lucie Mahaut, Christine Plumejeaud, Jonathan Storkey, Cyrille Violle, Sabrina Gaba

**Affiliations:** 1Laboratoire d’Écologie Alpine (LECA), Université Grenoble-Alpes, 2233 Rue de la Piscine, 38041 Grenoble Cedex 9, France; 2Anses, Laboratoire de la Santé des Végétaux, Unité Entomologie et Plantes Invasives, CBGP, 755 Avenue du Campus Agropolis, 34988 Montferrier-sur-Lez Cedex, France; guillaume.fried@anses.fr; 3FiBL, Research Institute of Organic Agriculture, 5070 Frick, Switzerland; larmengot@gmail.com; 4Agroécologie, AgroSup Dijon, INRAE, Université Bourgogne Franche-Comté, 21000 Dijon, France; berenger.bourgeois.1@ulaval.ca (B.B.); Lucie.MAHAUT@cefe.cnrs.fr (L.M.); 5Centre de Synthèse et d’Analyse sur la Biodiversité - Fondation pour la Recherche sur la Biodiversité, 13100 Aix-en-Provence, France; 6Département de Phytologie, Université Laval, Québec, QC G1V 0A6, Canada; 7Centre d’Etudes Biologiques de Chizé, CNRS & University La Rochelle, 79360 Villiers-en-Bois, France; Vincent.BRETAGNOLLE@cebc.cnrs.fr (V.B.); sabrina.gaba@inrae.fr (S.G.); 8LTSER Zone Atelier Plaine & Val de Sèvre, CNRS, F-79360 Villiers-en-Bois, France; 9INRAE GAFL UR 1052, Unité de Génétique et Amélioration des Fruits et Légumes, CS 60094, 84143 Montfavet Cedex, France; joel.chadoeuf@inrae.fr; 10UMR 5175 Centre d’Ecologie Fonctionnelle et Evolutive, Univ. Montpellier, CNRS, EPHE, IRD, Univ. Paul Valéry Montpellier 3, 34293 Montpellier, France; cyrille.violle@cefe.cnrs.fr; 11Littoral Environnement et Sociétés, 2 rue Olympe de Gouges, 17 000 La Rochelle, France; christine.plumejeaud-perreau@univ-lr.fr; 12Rothamsted Research, Harpenden, Hertfordshire, AL5 2JQ, UK; jonathan.storkey@rothamsted.ac.uk; 13USC Agripop, INRAE, Centre d’Etudes Biologiques de Chizé, F-79360 Villiers-en-Bois, France

**Keywords:** arable fields, species pool, specialization, open habitats, biodiversity decline, sampling strategies, life form

## Abstract

The definition of “arable weeds” remains contentious. Although much attention has been devoted to specialized, segetal weeds, many taxa found in arable fields also commonly occur in other habitats. The extent to which adjacent habitats are favorable to the weed flora and act as potential sources of colonizers in arable fields remains unclear. In addition, weeds form assemblages with large spatiotemporal variability, so that many taxa in weed flora are rarely observed in plot-based surveys. We thus addressed the following questions: How often do weeds occur in other habitats than arable fields? How does including field edges extend the taxonomic and ecological diversity of weeds? How does the weed flora vary across surveys at different spatial and temporal scales? We built a comprehensive dataset of weed taxa in France by compiling weed flora, lists of specialized segetal weeds, and plot-based surveys in agricultural fields, with different spatial and temporal coverages. We informed life forms, biogeographical origins and conservation status of these weeds. We also defined a broader dataset of plants occupying open habitats in France and assessed habitat specialization of weeds and of other plant species absent from arable fields. Our results show that many arable weeds are frequently recorded in both arable fields and non-cultivated open habitats and are, on average, more generalist than species absent from arable fields. Surveys encompassing field edges included species also occurring in mesic grasslands and nitrophilous fringes, suggesting spill-over from surrounding habitats. A total of 71.5% of the French weed flora was not captured in plot-based surveys at regional and national scales, and many rare and declining taxa were of Mediterranean origin. This result underlines the importance of implementing conservation measures for specialist plant species that are particularly reliant on arable fields as a habitat, while also pointing out biotic homogenization of agricultural landscapes as a factor in the declining plant diversity of farmed landscapes. Our dataset provides a reference species pool for France, with associated ecological and biogeographical information.

## 1. Introduction

Arable weeds are plants adapted to intense and recurrent anthropogenic disturbance in arable fields. Despite these specific environmental constraints, weeds do not constitute a set of plants with clearly defined ecological characteristics, so that even the definition of "weed" remains contentious [1,2,3]. One reason for this is that weed assemblages aggregate species from diversely cultivated as well as non-cultivated adjacent habitats (e.g., dry grasslands, riverbanks, sand dunes, etc.). Therefore, weed assemblages in arable fields include plant species also adapted to surrounding habitats [4]. In addition, a high spatiotemporal turnover of management practices results in differences between the regional species pool of weeds potentially occurring in arable fields (γ diversity) and the composition of assemblages observed at a given place and given time (α diversity) [5]. Clarifying the composition of the pool of weed species likely to establish in arable fields and their ecological diversity is of major importance to better understand and manage weeds.

Jauzein [6] defined a list of “messicole” weeds in France, including agrestal or segetal species whose life cycles mimics that of crop species, and which are expected to be more specialized to these crops. Beside agrestal weeds, many other plants can grow in cultivated areas, including casual weeds that are occasionally observed within arable fields, which considerably extend the ecological diversity and size of the potential weed flora (e.g., French flora of Jauzein [7], ~1400 taxa). Likewise, Metcalfe et al. [8] characterized contrasting groups of ‘resident’ and ‘transient’ weed species. Because they rely on arable cultivation for their persistence, the most specialized ‘resident’ agrestal weeds are more vulnerable to changing cultivation practices and to intensive management (e.g., herbicide spray [9]; tillage [10,11]). They are, thus, targeted for conservation actions [12,13] such as, e.g., the French “National Action Plan” [14]. Conversely, the less specialized, ‘transient’ weed taxa are also found in open habitats and are less submitted to agricultural constraints thanks to their extended niche. These generalist species are more likely to establish and persist in cultivated fields through repeated colonization despite changing management practices. A more generalist strategy should also be advantageous in mosaic landscapes including diverse and dynamic habitats. To monitor and forecast weed community dynamics, a comprehensive weed flora must incorporate both the few very specialized taxa, many of which are rare and declining, and the more opportunistic generalist taxa immigrating from habitats surrounding cultivated fields [15,16,17]. Different dynamics and conservation issues are expected for specialized and generalist weeds.

Another specific feature of weed assemblages is their high compositional variation in space and time, because of (i) source-sink dynamics between fields and the surrounding landscape [17,18,19], (ii) the history of anthropogenic introduction [20], and (iii) changing crop management practices over time [21]. We would expect, therefore, substantial differences between the composition of a comprehensive pool integrating the great ecological diversity of weeds at large scale and over a long term, and the composition of assemblages sampled over limited spatial and temporal extents [22]. The mismatch allows identification of the more instable and vulnerable species, and complements the analysis of ecological specialization of weed taxa to forecast weed dynamics. 

Here, we characterized (i) the ecological specialization of weeds (defined as species spontaneously growing in arable fields), and (ii) the compositional differences between a comprehensive weed flora and weed assemblages sampled in spatially and temporally restricted surveys. For (i), we examined whether the species present in a comprehensive flora of cultivated fields [7] were more generalist than species solely found in open herbaceous habitats. We used a large dataset of plant assemblages sampled in open herbaceous habitats (Divgrass initiative project [23]) to calculate the specialization of plants sampled across these habitats. We also addressed how weed diversity changes in the plot-based surveys including or not field-edge and margin (Biovigilance [24] and LTSER Zone Atelier “Plaine & Val de Sèvre”, hereafter ZA-PVS [25]), to examine any spill-over of generalist weeds occurring in surrounding habitats. For (ii), we compared the comprehensive weed flora to plot-based surveys with limited spatial and temporal extents. To discuss the factors underlying the differences and the consequences for conservation, we characterized compositional changes in terms of life forms, biogeographical origins, and conservation status. We performed the analyses in France, which incorporates a wide diversity of environmental conditions and agricultural contexts. We provide a comprehensive dataset with consistent and up-to-date taxonomic treatment [26,27], including biogeographical and ecological information. Bearing in mind the intrinsic ecological and biogeographical diversity of weeds, the dataset can be viewed as a reference species pool for ecological analyses of weed community assembly in temperate Europe.

## 2. Material and Methods

### 2.1. Datasets with Distinct Methodologies and Scopes

#### 2.1.1. Reference Lists and Flora

Expert knowledge datasets included (i) a reference list of specialized “messicoles” (i.e., agrestal or segetal) taxa specifically found in arable fields [14], and (ii) a comprehensive flora of cultivated fields over the whole of France [7], including plants also present in other habitats:

- The list of segetal species was determined based on a broad definition of “messicoles” [6]. In the classical definition, these weeds are related to cereal crops and should be monocarpic, annual, winter germinating. In a stricter definition, they should also be archaeophytes, introduced during early stage of agricultural development, i.e., before the Middle Ages in Europe. In an even stricter definition, they should have evolved morphologically and phenologically to mimic crops. Cambecèdes et al. [14] included species that are confined to farmlands in France and, more specifically, annual species (mostly germinating and emerging in autumn and in winter) occurring in winter cereal crops or other autumn-sown crops (e.g., oilseed rape). They also included geophytes with bulbs typically associated with crops, either cereals (*Bunium bulbocastanum*, *Gladiolus italicus*) or hoed crops (*Tulipa* spp.). 

- The comprehensive flora of Jauzein [7] included all species that can be found in France in fields where the soil is subjected to regular tillage, or fields under no-tillage practices but where weeding is still intense. Some grassland species able to survive in grass strips in the managed inter-row of perennial crops (vineyards and more generally orchards) were also included. However, the flora excluded plants growing in disturbed habitats but found only very casually in agricultural fields (apophytes). 

These two datasets were built on the long-term expertise of field botanists who explored countless arable fields over several centuries. 

#### 2.1.2. Plot-Based Community Sampling

We compiled weed assemblages sampled with standardized protocols [22], at the national scale (Biovigilance [24]), and at a regional scale in the west of France (LTSER Zone Atelier “Plaine & Val de Sèvre”, hereafter ZA-PVS [25,28]). Sampling plots were always located within arable fields, but margins (strips without crop near field boundary) were included or not depending on the objectives of the survey.

- In Biovigilance, between 268 and 814 fields were surveyed each year from 2002 to 2010. Data from a total of 5428 surveys over 1440 fields are available. Biovigilance covers almost all of France, with a stratified sampling in order to be representative of the main crop species and soil types in each region [24]. It includes 44 main crop species (winter cereals, 48%; maize, 21%; oilseed rape, 9%; sunflower, 6%), but focuses on main production areas excluding marginal production areas (mountains, Mediterranean area), thus, being representative of the most intensive farming practices. An area of 2000 m^2^ was surveyed twice a year within each field, at least 20 m away from field boundaries. We considered the list of weed species observed in plots treated before and after herbicide spraying, as well as in control plots without weeding.

- Around 3000 surveys were conducted between 2006 and 2016 in ZA-PVS area located in the west of France. The sampling protocol evolved through time, starting with weed sampling in a star-shaped array of 32 plots of 4 m^2^ (2 × 2 m) per field [28,29] to weed sampling in two transects of 10 plots of 1 m^2^ in the center of the field [17]. In addition to sampling at the center of the field, all field margins were sampled using transects (50 m), or five plots of 1 m^2^ depending on the year. The plots within fields were subjected to varying weeding management, from organic farming to chemical and mechanical weeding practices. The global dataset, thus, acknowledged the diversity of management practices in the area.

These lists and datasets differ in scope, methodology and objectives, but are among the most comprehensive vegetation surveys of arable floras available in France and represent the typical diversity of data available to characterize weed diversity in arable fields (Table 1). We updated taxonomic information using the TaxRef database version 10 [27], which complies with most recent French flora [26].

### 2.2. Biological, Ecological and Biogeographical Information

We obtained life form and biogeographical data from the Baseflor database [30]. Baseflor provides life form information following the classification of Raunkiaer [31]. Basic floristic zones represent the geographical extent and location of species, e.g., “eurasiatic” for species present in Europe and Asia, or “subtropical” for plants originating from subtropical areas. To investigate the conservation status of weeds, we collated data from the French Red list [32]. 

In order to characterize the ecological generalism of weeds, we used a database of ~96,000 surveys and 5245 plant taxa in open vegetation (i.e., without dominant cover by trees and shrubs) throughout France—the Divgrass database [23]. We analyzed the frequency of species co-occurrences within this dataset and identified groups of species co-occurring more often than expected by chance (modularity analysis [33]). We previously showed that the groups were related to different environmental conditions and that functional trait values varied across groups [23,34,35]. The groups, thus, correspond to distinct habitats, with varying taxonomic composition across groups and more consistent composition within groups. Specifically, we identified major grassland habitats, namely, dry calcareous grasslands, mountain grasslands, mesic grasslands, and ruderal and arable fields [23,34]. Other groups in the Divgrass database defined wetland, aquatic and ecotone habitats. Each species was assigned to a given main habitat but could also occur in sites of other habitats. The relative frequency of occurrence across habitats, the ‘coefficient of participation’ [36,37], quantified a degree of ecological generalism—the more often a species was found in other habitats than its main habitat, the more generalist it was [35].

### 2.3. Statistical Analyses

We compiled presence–absence information of weed taxa in the source datasets (Table 1), and analyzed taxa counts across categories of life form or of habitats by applying Chi-square tests. We compared the quantitative index of ecological generalism among groups of weeds by performing unpaired Wilcoxon tests. All statistical analyses were performed using the R software [38].

## 3. Results

Our compiled dataset included 1514 weed taxa in France (available on the Zenodo repository [39]), including 55 infra-specific taxa. The dataset indicates the data sources in which each weed taxon is found (Table 1). The most frequent families were Asteraceae (201), Poaceae (199), Fabaceae (184) and Brassicaceae (92), together representing 44.6% of the dataset. These families belong to the top six families in French flora, while Rosaceae and Orchidaceae do not rank as high in cultivation contexts than in the overall flora. In addition, there are 6060 plant species in France, of which 5351 are native. The compiled weed dataset, thus, included ca. 28.3% of the native French flora. Of these, 1402 taxa were included in the Jauzein flora (92.5% of the dataset). Our dataset displayed a broad diversity of life forms: 60.4% of species were therophytes, 25.7% were hemicryptophytes, 9.6% were geophytes and 1.7% were phanerophytes or chamaephytes.

### 3.1. Sampling Intensity and Spatial Coverage

The Biovigilance Flore Network reported a total of 332 taxa in 1440 fields throughout France (21.9% of the dataset). Although restricted to a 450 km^2^ area in Western France (<0.1% of the French territory), the ZA-PVS dataset surveyed weed assemblages in 3000 fields over 10 years and included 399 taxa of the dataset (26.3%) (Figure 1).

The Jauzein flora included 1003 taxa (66.2% of our weed dataset) that were absent from the two plot-based datasets (Figure 1). In addition, all specialized segetal species were included in Jauzein’s flora, but only 100 of them (38.7%) were recorded in ZA-PVS and Biovigilance Network plots. Conversely, plot-based datasets included 102 taxa (20.4% of these surveys) absent from Jauzein’s flora. Plot-based datasets, thus, only captured a small fraction of the overall flora, but also taxa not identified as typical agricultural weeds in weed flora. 

In total, 98 taxa from Jauzein [7] were included in the French Red List comprising 778 taxa—Data Deficient (DD) and Least Concern (LC) taxa being excluded (UICN France et al., 2012)—but only two, *Bupleurum subovatum* and *Nigella arvensis*, were present in the French plot-based surveys. 

### 3.2. Weed Preferred Habitats

The Divgrass survey characterized the main habitat of 5245 plant species in France. A total of 1248 (82.4%) weed taxa in our compiled weed dataset could be assigned to a main habitat from Divgrass, among which, 1161 were related to one of four major habitats (Table 2); namely, 706 were linked to ruderal and trampled grasslands (60.8%, including species-rich Mediterranean vegetation), 168 to mesic grasslands (14.5%), 162 to dry calcareous grasslands (14.0%), and 125 to mesophilous and nitrophilous fringes (10.8%). Taxa from dry calcareous grasslands were less frequent in the weed dataset than in the Divgrass set of species found in open habitats (13.0 vs. 20.5%, χ2 *p* < 0.001), suggesting that these taxa are less adapted to the context of cultivated fields (Table 2). Likewise, mountain grasslands were the second most important habitat in Divgrass (22.6% of taxa), but was associated with only 0.96% of weed taxa in our weed dataset. Conversely, species from mesophilous fringes and ecotones were more frequent among weeds than expected, based on proportions in the Divgrass database, especially for the plot-based ZA-PVS and Biovigilance datasets (Table 2). It could reflect the influence of hedges and vegetation surrounding arable fields. 

In terms of life form, most therophytes (75.6%) of our weed dataset were associated with ruderal and trampled grasslands, while 23.9% and 26.2% of hemicryptophytes were associated with mesic grasslands and ruderal habitats, respectively (Table 2). In total, 45.9% of chamaephytes and phanerophytes of our weed dataset were related to dry calcareous grasslands. 

### 3.3. Weed Habitat Specialization

Figure 2 compares the ecological generalism of weeds (present in our weed dataset) to that of other plants reported in Divgrass, for the four habitats including most weeds. For each habitat, a higher coefficient of participation (*c*) was found for weeds, which means that they could be found in a wider diversity of habitats, and thus, were on average more generalist than non-weeds found in those habitats (all Wilcoxon *p* < 0.001, Figure 2). In addition, weeds associated with ruderal and trampled grasslands in Divgrass were significantly more specialized compared to weeds related to other habitats (lower *c* values, Wilcoxon’s *W* = 198,377, *p* < 0.001). 

### 3.4. Biogeographic Origin 

The most frequent biogeographic origin status in our weed dataset was Mediterranean (602 taxa), followed with European (284), Eurasian (251), cosmopolitan (121) and introduced (105). Mediterranean taxa were more frequent among weeds than among other taxa of the Divgrass database (39.8 vs. 20.7%, χ2 *p* < 0.001). In total, 56.7% of weed taxa related to the ruderal habitat were of Mediterranean origin, while they were only 11.9% and 4% in mesic grasslands and mesophilous fringes, respectively. Almost half of the therophyte weeds (48.8%) were Mediterranean, while the proportion was 26.8% for other life forms. 

A lower proportion of weed taxa was of Mediterranean origin in the plot-based datasets (12.3% in Biovigilance and ZA-PVS), compared to Jauzein’s flora (42%). In total, 531 (52.9%) taxa present in Jauzein’s flora and absent from plot-based Biovigilance and ZA-PVS surveys were of Mediterranean origin, while only 58 (14.5%) species present in both the flora and plot-based surveys were of Mediterranean origin. Nevertheless, 58% of non-Mediterranean taxa in Jauzein were also not recorded in the plot-based surveys. 

In total, 65 out the 98 weed taxa found in the French Red List, DD and LC taxa excluded, were of Mediterranean origin (greater proportion than among weed taxa absent from the Red List, χ2 = 29.7, *p* < 0.001). Therefore, a great part of the current most threatened weeds in France are of Mediterranean origin. 

## 4. Discussion

Among the taxa found in open herbaceous vegetation in France (Divgrass database [23]), those that can occur in cultivated fields were more generalist that those not reported in cultivated fields. In addition, Jauzein’s flora, integrating long-term and broad-scale records, included far more taxa than more local and recent plot-based surveys of weed assemblages. A significant proportion of arable weeds were of Mediterranean origin, especially the taxa that are most threatened and vulnerable in France. Pervasive rarity could reflect the fact that current agricultural practices are less favorable for these weeds than ancient practices at the origin of their introduction in temperate Europe [40]. It could also reflect regional extinctions and environmental differences across regions, as well as a ‘dark diversity’ of weeds absent in local assemblages, but still regionally present [41]. Conversely, local plot-based surveys included taxa absent from Jauzein’s flora, but these taxa are probably occasional colonizers poorly adapted to the core of arable fields. The indeterminate nature of the weed species pool underlines the fuzzy limits of weed assemblages and the influence of surrounding habitats, providing opportunistic and casual immigrants. Weeds do not only include annuals restricted to arable fields, but also cover a broad spectrum of life forms and inhabit a range of non-cultivated habitats. Therefore, we propose that opposed to using the term ‘weed’ as a discrete categorization, ‘weediness’ is best viewed as a measure of specialization along an ecological continuum. 

### 4.1. Habitats of Agricultural Weeds

Almost all agricultural weeds can be found in other types of ecosystems [7]. Most weeds in our dataset (706) were associated with a broad habitat category of ruderal and trampled grasslands in Divgrass, encompassing other non-agricultural but still heavily disturbed environmental contexts. Conversely, 330 weed taxa were associated with permanent grassland habitats, e.g., dry calcareous and mesic grasslands. Do these non-cropped habitats represent a primary habitat of weeds, or are they secondarily colonized by weeds? Most species listed as segetal weeds [14] are by definition non-native weeds, introduced thousands of years ago in Europe (archaeophytes) as contaminants of cereal seeds [6]. This subset of weeds primarily occurred in arable fields in Western Europe, but some of them could also find suitable conditions in semi-natural habitats (e.g., dry open grasslands), acting as a refuge when weed control has become more intensive within fields. In fact, our results indicate that a significant proportion of segetal weeds are mainly associated to permanent herbaceous habitats (65, 28.9%, Table 2). Therefore, many segetal weeds are not confined to arable agricultural systems. It has been shown that segetal species with broader habitat preferences are less threatened [42,43]. Conversely, the species that are more specialized and less able to colonize habitats other than arable fields have been much impacted by agricultural intensification.

The ecological conditions of some habitats can favor species able to colonize and persist in arable fields (apophytes). For example, weeds characteristic of mesophilous and nitrophilous fringes (e.g., *Aethusa cynapium*, *Descurainia sophia*, *Galium aparine*) can well persist in fertilized arable fields thanks to their fast growth (through the increase in plant leaf area) in N-rich soils [42,43]. The vegetation in dry grasslands often remains short and sparse, offering numerous gaps where more stress-tolerant annuals with short life cycles can colonize and produce seeds before the summer drought. These annuals (e.g., *Iberis pinnata*, *Melampyrum arvense*, *Teucrium botrys*) can easily colonize stony fields. Plants of these non-agricultural habitats could have been selected for annual life cycle in relation to summer drought stress and were pre-adapted to the life cycle of autumn-seeded crops (e.g., *Cyanus segetum* [44]). They are especially present in Mediterranean vegetation, which is consistent with the great proportion of taxa of Mediterranean origin in our weed dataset.

### 4.2. Agricultural Weeds Tend to Be More Generalist

Only a very limited set of annual “messicole” weeds are exclusively found in crops (e.g., cereal mimetic species selected from wild relatives through seed sorting, such as *Bromus secalinus* or *Lolium temulentum*), and are targeted for conservation [14]. Conversely, we compared the ecological specialization of weeds and of other taxa in Divgrass (non-weeds present in open vegetation), and found consistently higher generalism of weeds for each of the main habitats (Figure 2). First, generalist weeds are likely to be more frequent in semi-natural habitats surrounding arable fields, and thus, to more easily enter weed assemblages in these fields. Mass effect and source-sink dynamics can then maintain weed populations in agricultural fields [8,16,17]. Second, more generalist taxa can show greater phenotypic and genotypic plasticity, enabling them to survive under specific environmental constraints of cultivated fields [45]. The index of generalism and the knowledge of the habitat spectrum of weeds should be useful for predicting the potential for immigration of weeds into cultivated fields from the surrounding habitat matrix. The probability of immigration could be weighted depending on the niche preferences and ecological generalism of weeds [46]. A perspective will be to further integrate an index of weed specialization depending on crop type [47].

### 4.3. Weed Assemblages at Field Margins Expand the Life Form Spectrum

Different sampling designs were used in plot-based datasets, with varying quadrat size (1 to 4 m^2^ in ZA-PVS, 2000m^2^ in Biovigilance), number of quadrats (20 to 32 in ZA-PVS, 2 in Biovigilance), number of replicates per year (1 in ZA-PVS, two in Biovigilance), and including (ZA-PVS) or not (Biovigilance) field margin. The different designs directly affected the number of weed taxa recorded, since (i) higher sampling effort increases the detection of rare species, (ii) species richness and the presence of threatened weeds are higher in field margins [8,17,48], and (iii) field margins are likely to shelter casual taxa from adjacent habitats. Including field margins enlarges the taxonomic and ecological diversity of weed taxa in the plot-based surveys [49,50]. The relative number of species from permanent grasslands was, thus, higher in the ZA-PVS dataset including field margins than in Biovigilance only including plots far from field edge (Table 2).

It is generally claimed that arable weeds are almost exclusively annual plants [6]. Although therophyte was the dominant life form (58.4%), our weed dataset encompassed a broad biological spectrum, even including some phanerophytes and chamaephytes. Geophytes (9.5%) and therophytes represent strategies most tolerant to regular disturbances. The proportion of therophytes was between 80 and 90 %, and together with geophytes, 94% of individuals in the sampling plots of Biovigilance [48], but the two types only represented 67.9% of species in our dataset. It indicates that therophyte and geophyte remain dominant strategies within fields, while other types can be present but are less abundant. In addition, although hemicryptophytes are more dominant in herbaceous habitats adjacent to the fields, they can secondarily be therophytes (biannuals, e.g., *Chondrilla*) or geophytes (*Rumex*), depending on a balance between sexual and vegetative reproduction [7].

Woody perennials are usually introduced in arable fields by wind (*Acer*, *Clematis*, *Fraxinus*) or by animals (*Quercus*, *Rosa*, *Sambucus*). They are casual and their persistence within fields is not compatible with the intensity of disturbance associated with field management. All the woody perennials (41 taxa) were absent from Jauzein’s flora but seedlings were regularly found in the plot-based surveys, especially the ZA-PVS dataset including samples at field margin. Furthermore, herbaceous taxa absent from Jauzein’s flora were associated with adjacent and ecotone habitats (hedges, ditches, grasslands, woods), and were, thus, not recognized as a typical weed in the French flora, such as *Arctium lappa*, *Brachypodium sylvaticum* or *Stellaria holostea*. An interesting avenue of future research would be to more clearly define the proportional contribution of these transient species to the assembly and functioning of the weed flora in contrasting landscapes.

### 4.4. Mediterranean Origins and Conservation Issues 

We found substantial mismatch between species lists derived from the plot-based surveys and the comprehensive Jauzein’s flora. A total of 71.5% of the species present in Jauzein’s flora were not captured in Biovigilance and ZA-PVS plot-based surveys, a majority of which (52.9%) were of Mediterranean origin. First, this can be explained by the limited spatial coverage of the plot-based surveys. ZA-PVS is located in the Western part of France and Biovigilance has been set up in the main agricultural basins of France, hence, it is representative of the dominant intensive agriculture systems, excluding marginal farming systems. Therefore, a lot of common weeds in the Mediterranean (e.g., *Anisantha madritensis*, *Erodium malacoides*, *Medicago orbicularis*) were absent from the two plot-based surveys. Second, it is unlikely that two surveys representing 4500 fields (ca. 40,000 ha) can detect the whole pool of weeds present over 18.4 million ha (the surface or arable land in France), with some species being intrinsically rare and/or spatially localized. Mediterranean weeds were regularly reported in the fields of North France in ancient flora, although many have not been reported more recently [14]. Third, the absence of information on the seed bank can also explain the discrepancy. Weeds generally have persistent seed banks and only part of it is expressed as flora during a cropping season [51]. Although most weed diversity could be present in the seed bank, the temporal extent of weed monitoring in the Biovigilance network (9 years) and of ZA-PVS (10 years) still makes this hypothesis unlikely. 

Fourth, the population dynamics and abundance of weeds have much changed in space and time. High rarity and rapid decline of some segetal species has motivated specific conservation policies for this group of species [52]. Only 100 of the segetal taxa mentioned in Cambecèdes et al. [14], hence 38.7%, were recorded in the plot-based surveys, suggesting that these weeds could be on the verge of extinction in major French arable basins (North, Centre, West). Intensification of agricultural practices and in particular, the intensive use of herbicides, is mainly responsible for the decline of both rare [9] and common weeds [53]. The deficit of weed taxa in the plot-based surveys can, thus, reveal the decline of local weed diversity, since the time period between the comprehensive flora (1995) and the plot-based survey (from 2002 to 2016) ranged from 7 to 21 years. A recent meta-analysis showed a decline of 20% of weed taxa in Europe, mostly before the 1980s [54]. Yet, many of these species can still be found on unfertile calcareous soils in the south of France, where traditional farming systems with low inputs are maintained [52]. 

While a significant part of arable weed diversity in France [55] and in Europe [13] is of Mediterranean origin, most of these species are increasingly rare and most often undetected in recent vegetation sampling in arable fields. A substantial proportion of weeds present in the French Red List are of Mediterranean origin. There is urgent need of monitoring and research programs in weed-rich areas, such as southern Europe, in order to update the Red Lists and to design efficient conservation measures.

## 5. Conclusions and Perspectives

The weed flora is of high interest because of its potential impact on crop production, but also because of its importance for biodiversity conservation [56] and provision of ecosystem services [57,58,59]. Contrary to common thinking, our results indicate that weeds cover wide ecological and biogeographical spectra, and only 58.4% of weeds in our comprehensive dataset are therophytes. Apart from typical annual weeds, arable fields also provide a habitat for generalist plants found across a range of herbaceous habitats. The diverse ecology of weeds can be related to the spatiotemporal variability of environmental conditions in arable fields. Species mostly occurring in permanent grasslands can disperse from surrounding habitats in mosaic landscapes. They can also establish within fields under particular farming practices, such as no-till or heavily fertilized plots (species from nitrophilous fringes), or under specific abiotic conditions such as shallow calcareous clay soil (species from dry calcareous grassland). To better integrate the ecological diversity of weeds in studies of weed population and community dynamics, we proposed an index of habitat specialization that can serve as a flexible measure of weediness in place of a too sharp binary categorization of weeds. A weediness index can also integrate species characteristics such as functional traits, allowing greater weed frequency and performance in agricultural fields [1,60]. 

Despite a broad spatial coverage, we found that weed assemblages sampled in mostly intensive cropping systems of the main cereal plains of France include only a small fraction of the potential weed species pool (~34%). Landscape and biotic homogenization greatly alter the persistence and diversity of weeds [61], and our results underline the vulnerability and limited viability of specialized segetal species. Great species impoverishment in observed assemblages [41] should be acknowledged to properly characterize the drivers of weed community dynamics [62]. Because of the complexity of these dynamics in space and time [17,63], plot-based community composition at a given time should be analyzed and compared to the composition of a wider pool of potentially present weeds. The pool should not only include the bulk of sampled species, but also, according to the study aims: (i) species from adjacent habitats that can reach a local community through dispersal (e.g., tree seedlings), (ii) species in the seed bank that can coexist in the target community, or (iii) locally extinct species that were known to occur previously (diachronic studies). Weed ecology provides a flagship case for addressing the sustainability of complex metacommunity dynamics in heterogeneous agricultural landscapes [64]. Using our dataset with ecological and biogeographical information should provide a powerful tool for delineating weed species pools and designing appropriate statistical analyses [65]. 

## Figures and Tables

**Figure 1 plants-09-00824-f001:**
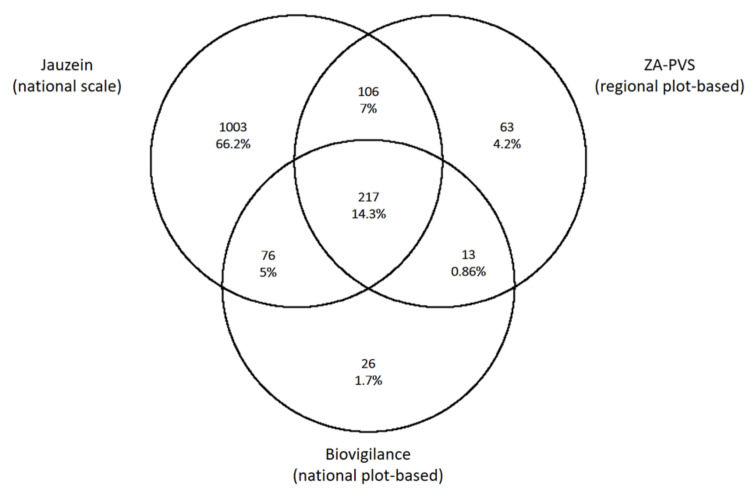
Venn diagram of species numbers in (i) Jauzein’s French flora, (ii) Biovigilance National plot-based survey in France, (iii) ZA-PVS regional plot-based survey in Western France.

**Figure 2 plants-09-00824-f002:**
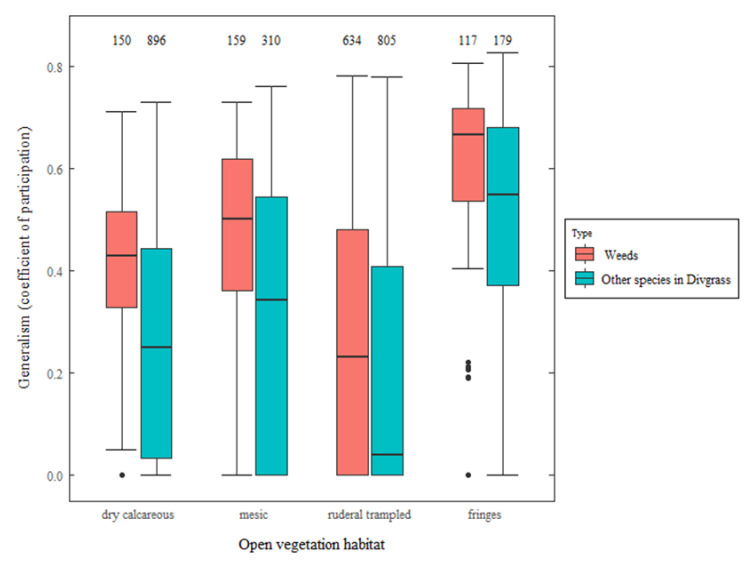
Network-based index of ecological generalism (coefficient of participation, denoted *c*) based on the Divgrass dataset of open vegetation in France [23]. Boxplots represent the variation of generalism of species belonging to the four major habitats of Table 2 (abscissa), for weeds (red) and for other plant species found in Divgrass (blue). Sample sizes are shown above the boxplots. The Wilcoxon statistic of comparison between weeds and other species in Divgrass is *W* = 93,930 for habitat 1, *W* = 32,873 for habitat 3, *W* = 309,708 for habitat 5, and *W* = 13,029 for habitat 9. All the Wilcoxon tests are significant (*p* < 0.001).

**Table 1 plants-09-00824-t001:** Datasets considered in the present study, with corresponding methodological information, spatial and temporal extents, and species numbers.

Dataset	Methodology	Spatial Extent	Temporal Extent	Species Number	References
Flora of cultivated fields	All wild plant taxa reported in cultivated fields	Whole France	Unlimited	1402	[7]
National “messicole” list	Plant taxa reputed to be specific to crops	Whole France	Unlimited	258	[14]
Divgrass	Phytosociological surveys, with classes of species abundances	Whole France	Unlimited	5245	[23]
Biovigilance Flore network	2000 m^2^ quadrats, 1440 arable fields (core of fields)	Whole France	9 years (2002–2010)	332	[24]
LTSER Zone Atelier “Plaine & Val de Sèvre” (ZA-PVS)	20 to 32 sampling plots 1 to 4 m^2^, c.200 fields per year (core of fields plus margin)	West of France (450 km^2^)	10 years (2006–2016)	399	[25]

**Table 2 plants-09-00824-t002:** Habitat types to which most weed taxa (white columns) and non-weed taxa (grey column, Divgrass database) belong. The habitats were derived from an analysis of species co-occurrences in the Divgrass database of open vegetation in France. The expected numbers of taxa for each list of weed taxa were calculated based on the proportions of taxa across habitats in Divgrass dataset (grey column, italic). When comparing species counts in Divgrass to the counts in each list, the difference of proportions across habitats is always significant (χ2 tests, all *p* < 0.001).

		Observed (and Expected) Number of Taxa				
	*Divgrass*	Agrestal taxa	Jauzein	Biovigilance	ZA-PVS	**Global dataset**
Dry calcareous grasslands	*1076*	27 (73)	131 (353)	21 (96)	56 (119)	**162 (380)**
Mesic grasslands	*472*	27 (32)	152 (155)	63 (42)	102 (52)	**168 (167)**
Ruderal and trampled grasslands	*1447*	160 (99)	691 (474)	160 (129)	148 (160)	**706 (510)**
Mesophilous and nitrophilous fringes	*296*	11 (20)	105 (97)	50 (26)	55 (33)	**125 (104)**

## Data Availability

The complete dataset of weed taxa with occurrence information in the data sources (Table 1) and ecological and biogeographical data is available on https://zenodo.org/record/1112342.

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
