# Peer review of "Ecological Specialization and Rarity of Arable Weeds: Insights from a Comprehensive Survey in France"

_plants, 2020, doi:10.3390/plants9070824_

Round 1

Reviewer 1 Report

I had the impression that your interesting article combines two important topics: preserving biodiversity and management of weeds.

The Abstract is a bit long, but informative and creates interest in the reader. The main points of the complete article are well represented within the Abstract.

The Introduction is appropriate. It gives a solid background for the investigation. Objectives are well described. I would have liked to have some of the specific terms (such as “ecological amplitude” or “infra-taxa”) explained.

In line 111, “global list” is misleading. If I understand right, this list was created from data obtained in France only, not from all over the world.

The “Materials and Methods” describe the databases, but later on during the manuscript I find that some details are missing (see my comments for the Results).

Results: The presentation of results are presented Figure 1 is very informative.

A little technical remark: line numbering starts again (from 1) after Tables, and I don’t think it should be this way.

What is the definition of “expected number of species”, as referred to in Table 2? I think you may have to provide more information on the Divgrass database, and on how this expected number was calculated.

Conclusions are well grounded.

I would like to clarify a small observation here: What do you mean by “Species shared with permanent grasslands”, in line 179? I think you mean species most often found in permanent grasslands.

Reference No 2, 6, 10, 22, 34 and 65 have different styles than the others.

Author Response

Review 1

Comments and Suggestions for Authors

I would have liked to have some of the specific terms (such as “ecological amplitude” or “infra-taxa”) explained.

We have changed “ecological amplitude” to “ecological generalism”. “infra-taxa” represent all taxa defined below species level (subspecies, variety…). We have changed it to “infra-specific taxa” for greater clarity.

In line 111, “global list” is misleading. If I understand right, this list was created from data obtained in France only, not from all over the world.

We agree, and have changed the wording into “(our weed) dataset”.

A little technical remark: line numbering starts again (from 1) after Tables, and I don’t think it should be this way.

It did not appear so on my computer. We hope there will no longer be such issue with the newly submitted manuscript.

What is the definition of “expected number of species”, as referred to in Table 2? I think you may have to provide more information on the Divgrass database, and on how this expected number was calculated.

We have modified the caption of Table 2 to clarify this point: “The expected numbers of taxa for each list of weed taxa were calculated based on the proportions of taxa across habitats in Divgrass dataset (grey column, italic).”

I would like to clarify a small observation here: What do you mean by “Species shared with permanent grasslands”, in line 179? I think you mean species most often found in permanent grasslands.

We have changed the wording to “Species mostly occurring in permanent grasslands”.

Reference No 2, 6, 10, 22, 34 and 65 have different styles than the others.

We have checked and homogenized the formatting of references.

Reviewer 2 Report

In general the article is well written, while I consider that the manuscript contains information that deserve to be published after minor revision. The main disadvantage of this paper is that the authors they didn’t provided information about the weed species that recorded in this study. I provide below a few suggestions that, if the authors decide to implement into the paper, the paper will improved.

Comments

Main suggestions

  1. Authors should add a table presented the number of species per family that recorded in this survey.
  2. Tables about the most common weeds (e.g. grass weeds, broadleaved weeds, perennial weeds and biennial weeds) that recorded in this survey should be added. Authors should add 10 to 20 weeds in each table. These data is necessary in order to show the diversity in France and in the Mediterranean region around the fields.
  3. More information about the period tha the authors make the measurments should be presented

This information is necessary in order the authors to support their results.

Minor corrections

Line 151: 2000 m2 should be corrected as 2000 m2

The authors should used a english word for the word releves (line 173 and table 1)

Line 277: Parenthesis is missing in 2009. The authors should correct the references.

Author Response

Review 2

Comments and Suggestions for Authors

Authors should add a table presented the number of species per family that recorded in this survey.

We have indicated at the beginning of Results the number of taxa in most common families: “The most frequent families were Asteraceae (201), Poaceae (199), Fabaceae (188) and Brassicaceae (95), together representing 44.6% of the list. These families belong to the top six families in French flora, while Rosaceae and Orchidaceae do not rank as high in cultivation contexts than in the overall flora.”

Tables about the most common weeds (e.g. grass weeds, broadleaved weeds, perennial weeds and biennial weeds) that recorded in this survey should be added. Authors should add 10 to 20 weeds in each table. These data is necessary in order to show the diversity in France and in the Mediterranean region around the fields.

We do not have information on weed commonness for the data taken from flora and species lists (Jauzein, 1995; Cambecedes et al., 2012). Plot-based data provide information on local and regional abundances, and we work currently on a follow-up paper that will provide further insights on the drivers of weed commonness and rarity in France and UK.
In addition, the biological traits of specialized vs generalist weeds have been examined in another recent paper, Bourgeois, Bérenger, François Munoz, Guillaume Fried, Lucie Mahaut, Laura Armengot, Pierre Denelle, Jonathan Storkey, Sabrina Gaba, et Cyrille Violle. 2019. « What makes a weed a weed? A large-scale evaluation of arable weeds through a functional lens ». American Journal of Botany 106 (1): 90‑100. https://doi.org/10.1002/ajb2.1213.

More information about the period that the authors make the measurements should be presented

We have included the years of sampling in Table 1.

Minor corrections

Line 151: 2000 m2 should be corrected as 2000 m2

Corrected.

The authors should used a english word for the word releves (line 173 and table 1)

We have changes “relevés” into “surveys”.

Line 277: Parenthesis is missing in 2009. The authors should correct the references.

We have checked the references.

Reviewer 3 Report

General comments

This research aims to provide information to understand why plants referred as weeds, those that can appear in arable fields, can be more or less abundant based on a database created by the authors. This database is constructed on three surveys data sources from France that cover the whole country for a wide time span (around a decade), and including 44 main crops and a very good representation of non-crop habitats. Easily informing about the occurrence of each plant taxa in these data sources, habitats occupied, life forms and some other characteristics authors perform statistical analyses to understand why some species can be more generalist than others, i.e. rare arable plants only found in arable fields that are very specialized. Also interesting results are provided to understand how surrounding habitats in arable fields can contribute to the weed flora found in surveys, and their importance as arable species pool. Finally, a weediness index is proposed to be used in future studies, and also the data used is fully online available, and could be the base of further research. An important outcome is that a set of taxons of Mediterranean origin are specialist of arable fields and found nowhere else. This group of segetal species are under threat across Europe, and considering their higher occurrence in Southern Europe, conservation measures in these countries should be undertaken in arable fields.

The paper is very well written, methods correct, and results well-presented and clearly discussed. Therefore, this research fits within the scope of this Special Issue and merits publication after a minor revision is performed.

Please find below some comments to improve the manuscript.

For research articles with several authors, a short paragraph specifying their individual contributions must be provided at the end of the manuscript. Also, no details of funding are provided; if there are, please provide the details. Finally, please if there any Conflicts of Interests.

Minor comments

Introduction

Line 53: rewrite as: “… growing in arable fields adapted to their intense and …”.

Line 62: can you provide a reference to support this statement?

Line 71: or Line 73: The recent research by Torra et al. (2018) also supports the importance of tillage in explaining rare arable plants persistence. In either lines it could be added. Torra J, Recasens J, Royo-Esnal A (2018) Seedling emergence response of rare arable plants to soil tillage varies by species. PLOS ONE 13(6): e0199425.

Line 76: Andreasen et al. 1996 and Storkey et al. 2012 are not in the reference list.

Line 87: Please check the journal style regarding references into the text and follow it in a consistent manner.

Line 109: Gargominy 2016 is not in the reference list.

Material and Methods

Line 151: superscript for square meters

Line 156: please add spaces after 2.

Line 157: please add a space after 1. Also in line 159.

Line 158: please add a space after 50.

Line 158: change subject for subjected.

Results

Page 6 line 3: Probably I am missing something. According to the csv file, there are 1577 taxa in total. This file has 1578 files, and excepting the first one, the rest are plant taxons. Please review and verify the numbers provided.

Page 6 line 24: Jauzein (2005) is not in the reference ist.

Page 6 line 32: “… could be assigned to a main habitat ….”.

Discussion

Page 10 line 48: Partel (2011) is not in the reference list.

Page 11 line 11: Fried et al. (2010) is not in the reference list.

Page 12 line 172: “… but also because of its …”.

Page 13 line 203: Please add a dot at the end of the sentence.

References

Line 251: Authors’ names should be in capital letters according to the format of the other references.

Line 288: Authors’ names should be in capital letters according to the format of the other references. This reference is not used in the text.

Line 379: Authors’ names should be in capital letters according to the format of the other references. Please check the rest of references in the list.

Figure 1: I recommend including below the numbers the % they represent from the global list.

Figure 2: I recommend numbering the habitats from 1 to 4, not 1,3, 5 and 9.

Author Response

Review 3

For research articles with several authors, a short paragraph specifying their individual contributions must be provided at the end of the manuscript. Also, no details of funding are provided; if there are, please provide the details. Finally, please if there any Conflicts of Interests.

We have added an Author contributions section at the end of the main text, before the references. The Acknowledgement section includes information on funding.

Minor comments

Line 53: rewrite as: “… growing in arable fields adapted to their intense and …”.

We have reworded the sentence.

Line 62: can you provide a reference to support this statement?

We have included here a reference to Poggio, Santiago L., Enrique J. Chaneton, et Claudio M. Ghersa. 2010. « Landscape complexity differentially affects alpha, beta, and gamma diversities of plants occurring in fencerows and crop fields ». Biological Conservation 143 (11): 2477‑86

Line 71: or Line 73: The recent research by Torra et al. (2018) also supports the importance of tillage in explaining rare arable plants persistence. In either lines it could be added. Torra J, Recasens J, Royo-Esnal A (2018) Seedling emergence response of rare arable plants to soil tillage varies by species. PLOS ONE 13(6): e0199425.

Thank you for the suggestion. We have added this reference.

Line 76: Andreasen et al. 1996 and Storkey et al. 2012 are not in the reference list.

Thank you for the notice. We have added the missing item Andreasen et al. (1996) in the reference list, and corrected Storkey et al. (2011) into Storkey et al. (2012).

Line 87: Please check the journal style regarding references into the text and follow it in a consistent manner.

We have checked and corrected the format when needed.

Line 109: Gargominy 2016 is not in the reference list.

We have corrected into Gargominy et al. (2016).

Material and Methods

Line 151: superscript for square meters

Corrected.

Line 156: please add spaces after 2.
Line 157: please add a space after 1. Also in line 159.
Line 158: please add a space after 50.

Done.

Line 158: change subject for subjected.

Done

Results

Page 6 line 3: Probably I am missing something. According to the csv file, there are 1577 taxa in total. This file has 1578 files, and excepting the first one, the rest are plant taxons. Please review and verify the numbers provided.

Thank you for the notice. There are 1577 taxa in the Zenodo dataset, including data on British weeds that do not consider in the present paper. 1514 of these taxa are present in French dataset, and we have corrected the number in main text accordingly.
In addition we should update the version on Zenodo soon to remove Bupleurum lancifolium auct., because it should be considered a synonym of B. subovatum.

Page 6 line 24: Jauzein (2005) is not in the reference list.

We have corrected the reference to Jauzein (1995).

Page 6 line 32: “… could be assigned to a main habitat ….”.

Corrected.

Page 10 line 48: Partel (2011) is not in the reference list.

Corrected to Partel et al., 2011.

Page 11 line 11: Fried et al. (2010) is not in the reference list.

We have added this reference in the reference list.

Page 12 line 172: “… but also because of its …”.

Corrected.

Page 13 line 203: Please add a dot at the end of the sentence.

We have not found this sentence with a missing dot.

Line 251: Authors’ names should be in capital letters according to the format of the other references.
Line 379: Authors’ names should be in capital letters according to the format of the other references. Please check the rest of references in the list.

We have checked the references and corrected them when needed.

Line 288: Authors’ names should be in capital letters according to the format of the other references. This reference is not used in the text.

We have not found the reference. Maybe some issue with line numbering in the submitted manuscript?

Figure 1: I recommend including below the numbers the % they represent from the global list.

We have includes percentages accordingly.

Figure 2: I recommend numbering the habitats from 1 to 4, not 1,3, 5 and 9.

We have replaced the numbers with habitat names.